# Development of Bio-Functionalized, Raman Responsive, and Potentially Excretable Gold Nanoclusters

**DOI:** 10.3390/nano11092181

**Published:** 2021-08-25

**Authors:** Ryan D. Mellor, Andreas G. Schätzlein, Ijeoma F. Uchegbu

**Affiliations:** School of Pharmacy, University College London (UCL), 29–39 Brunswick Square, London WC1N 1AX, UK; ryan.mellor.16@ucl.ac.uk (R.D.M.); a.schatzlein@ucl.ac.uk (A.G.S.)

**Keywords:** ultrasmall-in-nano, excretable, gold, Raman, theranostic

## Abstract

Gold nanoparticles (AuNPs) are used experimentally for non-invasive in vivo Raman monitoring because they show a strong absorbance in the phototherapeutic window (650–850 nm), a feature that is accompanied by a particle size in excess of 100 nm. However, these AuNPs cannot be used clinically because they are likely to persist in mammalian systems and resist excretion. In this work, clustered ultrasmall (sub-5 nm) AuNP constructs for in vivo Raman diagnostic monitoring, which are also suitable for mammalian excretion, were synthesized and characterized. Sub-5 nm octadecyl amine (ODA)-coated AuNPs were clustered using a labile dithiol linker: ethylene glycol bis-mercaptoacetate (EGBMA). Upon clustering via a controlled reaction and finally coating with a polymeric amphiphile, a strong absorbance in the phototherapeutic window was demonstrated, thus showing the potential suitability of the construct for non-invasive in vivo detection and monitoring. The clusters, when labelled with a biphenyl-4-thiol (BPT) Raman tag, were shown to elicit a specific Raman response in plasma and to disaggregate back to sub-5 nm particles under physiological conditions (37 °C, 0.8 mM glutathione, pH 7.4). These data demonstrate the potential of these new AuNP clusters (Raman NanoTheranostics—RaNT) for in vivo applications while being in the excretable size window.

## 1. Introduction

Gold nanoparticles exhibit surface plasmon resonance, meaning that at a particular wavelength, dependent on the particles’ size and shape, they display increased absorbance [1,2]. This SPR band may be exploited by techniques such as Surface-Enhanced Raman Spectroscopy (SERS) and photothermal therapy [2]. In addition to the aforementioned SERS, used for the quantitative detection of organic compounds within biological matrices [3], there are several Raman-based techniques that can provide non-invasive insight into a compound’s immediate environment and be used for biomedical applications. Notably, Spatially Offset Raman Spectroscopy (SORS) [4] implements a spatial separation on a sample’s surface between the excitation source and the collection Raman zone, thus allowing for the subsurface detection and depth determination of analytes. Temperature Spatially Offset Raman Spectroscopy (T-SORS) [5,6] may be used to determine the temperature of media based on a measured anti-Stokes/Stokes ratio. These techniques provide detailed information about the physical and chemical environment of the nanoparticles and, by extension, may be used to provide information on the location of a Raman reporter in biological matrices [4] and the temperature of a biological microenvironment [5,6]. This information is currently unobtainable by non-invasive approaches; as such, the use of Raman-based techniques could be applied in the diagnosis of disease.

Gold nanoparticles have been extensively used for biomedical applications, and they have received ever-growing interest in recent years [2,7]. They have been employed as sensors [8,9], drug-delivery vehicles [10,11,12], and photothermal [13,14]/photodynamic [15,16] agents. For use in biomedical applications, gold nanoparticles must be excretable via the kidneys when parenterally injected, and the SPR band should reside in a region known as the phototherapeutic window (from 650 to 850 nm), which is a range of low absorbance by human tissue that results in a high depth penetration of an incident Raman beam [17]. Unfortunately, gold nanospheres that exhibit this SPR band tend to be in the size range of 100–400 nm [18], a particle size that cannot be sufficiently excreted by the kidneys [19]. Though gold nanorods do display SPR bands in the phototherapeutic window [20], there is no evidence that they can be manufactured in a renally-excretable size range. This is because the length of the rod acts to increase the hydrodynamic radius [21]. Particles that are excretable (sub-5 nm) [22,23] have SPR bands in the 515–520 nm wavelength range [18] and are thus not useful for non-invasive detection applications because the incident light would be strongly absorbed by tissue. The aim of this study was to synthesize labile [24] 100–400 nm clusters of nanoparticles from smaller sub-5 nm building blocks so that they could display the desired absorbance in the phototherapeutic window while being able to be safely excreted by the body.

Though gold itself is inert and biocompatible, problems may arise when considering the persistence of AuNPs within mammalian systems [25]. The body’s main mechanism for the excretion of compounds from circulation, namely via the renal pathway, is not efficient at removing particles larger than 5 nm in diameter [19]. This is mainly due to the functional pore size of the glomerular capillary wall of 4.5–5 nm [22]. Therefore, clustering must be reversible to allow for the regeneration of the initial sub-5 nm cores. Additionally, the 100-400 nm clusters of said gold nanoparticles, intended to elicit absorbance in the phototherapeutic window, should also allow for the passive targeting of tumours by the enhanced permeability and retention (EPR) effect and consequent tumoricidal activity [26].

For the first time, such a construct, termed a Raman NanoTheranostic (RaNT), was formed (Figure 1). The construct consists of sub-5 nm gold nanoparticles formed into a reversible 100–400 nm cluster, labelled with a Raman reporter (BPT), and coated with an amphiphilic polymer—N-palmitoyl-N-monomethyl-N,N-dimethyl-N,N,N-trimethyl-6-O-glycolchitosan (GCPQ) [27].

## 2. Materials and Methods

### 2.1. Materials

All the chemicals used in this study are listed in Appendix A.

### 2.2. Particle Synthesis

#### 2.2.1. Synthesis of Octadecyl Amine-Coated Gold Nanoparticles

Octadecyl amine (ODA) AuNPs were synthesized using sodium borohydride (NaBH_4_) as a reducing agent and ODA as a capping agent. This method produced small, organic-soluble AuNPs with a ligand that may be readily displaced by a thiol compound due to the relatively weak amine–gold interaction. The method was adapted from the work of Chen et al. [28].

Briefly, an HAuCl_4_.3H_2_O solution (3.94 mg mL^−1^ in chloroform; 10 mL) was added to an ODA solution (3 mg mL^−1^ in chloroform; 90 mL) and bath-sonicated for 30 min while protected from light. Under vigorous magnetic stirring, an NaBH_4_ solution (3.78 mg mL^−1^ in ethanol; 10 mL) was added dropwise and again bath-sonicated for 30 min while protected from light. The volume was reduced to ~2.5 mL by rotary evaporation. ODA AuNPs were precipitated by the addition of ethanol (200 mL) and stored at −20 °C overnight. The NPs were collected by vacuum filtration through a 0.2 µm nylon filter, washed with ethanol (100 mL), and dried under reduced pressure.

#### 2.2.2. Optimization of Clustering Reaction Conditions

Reaction conditions were optimized for ethylene glycol bis-mercaptoacetate (EGBMA) and ethanol concentrations. An ODA AuNP solution (1 mg mL^−1^, chloroform; 0.5 mL) was diluted with volumes of ethanol and chloroform so that the final reaction volume was 5 mL and the ethanol concentration varied from 0 to 10% *v*/*v*. An EGBMA solution (5 µL mL^−1^ in chloroform) was added, while bath-sonicating, to the ODA AuNP solution in volumes such that the final EGBMA concentration varied from 0 to 1 µL mL^−1^ (actual used volumes can be found in Appendix A). The solution was allowed to react for 24 h at room temperature. During this time, the clusters settled to the bottom of the reaction vessel. The pellet was collected by the decantation of the supernatant, and the clusters were resuspended in chloroform (0.5 mL).

#### 2.2.3. Transfer of Clusters into the Aqueous Phase

Clusters were prepared in chloroform to prevent the premature hydrolysis of the EGBMA linker. However, for the clusters to be of biological relevance, they had to be made water dispersible. This was achieved using GCPQ as a phase transfer agent. A GCPQ dispersion (0.1–10 mg mL^−1^ in methanol; 1 mL) was added to the EGBMA clusters (as resuspended in chloroform; 0.5 mL), and the solution vortexed for 10 s and bath-sonicated for 30 s. The two solvents—chosen for their miscibility, similar boiling points, and ability to dissolve their respective solutes—were removed by rotary evaporation. The thin film was hydrated using MilliQ water (mqH_2_O; 0.5 mL), vortexed for 30 s, and finally freeze-dried for storage.

#### 2.2.4. Biphenyl-4-Thiol Labelling of the Clusters

BPT (1 mg mL^−1^ in chloroform; 0.5 mL) was rapidly added to the ODA AuNPs (A450 = 0.54 in chloroform; 4 mL) 30 min before, concurrently, or 30 min after the addition of EGBMA (1 µL mL^−1^ in chloroform; 0.5 mL). The solution was allowed to react for 1 h at room temperature, and the clusters were allowed to settle overnight. The volume was reduced to 0.5 mL via the decantation of the supernatant. Subsequently, a GCPQ solution (2 mg mL^−1^ in methanol; 0.5 mL) was added, and then the liquid was vortexed for 10 s and bath-sonicated for 30 s. The solvent was removed by rotary evaporation. mqH_2_O (0.5 mL) was then used to hydrate the thin film, followed by vortexing for 30 s, and the dispersion was finally freeze-dried for storage.

#### 2.2.5. Optimization of Biphenyl-4-Thiol Surface Coverage

A BPT solution (0.8, 1.6, 2.4, 3.2, or 4.0 mg mL^−1^, chloroform; 0.5 mL) was rapidly added to previously synthesized EGBMA clusters obtained using the procedures described in Section 2.2.2 (resuspended in chloroform; 0.5 mL). The solution was vortexed for 10 s and bath-sonicated for 5 min, and the BPT-labelled clusters were allowed to settle overnight. The supernatant containing free BPT was removed and discarded. The pellet was resuspended (in chloroform; 0.5 mL), a GCPQ dispersion (5 mg mL^−1^ in methanol; 0.5 mL) was added, and the solution was vortexed for 10 s and bath-sonicated for 30 s. The solvent was removed by rotary evaporation. The thin film was then rehydrated (mqH_2_O; 0.5 mL), followed by vortexing for 30 s. The samples were analysed by Raman spectroscopy, and the intensity at 1585.8 cm^−1^ (ring stretching) was recorded.

### 2.3. Particle Characterization

#### 2.3.1. UV–Vis Spectroscopy

UV–Vis spectra were recorded on undiluted aqueous cluster samples, prepared as described above, from 350 to 1100 nm and a step size of 0.5 nm (UV-1650PC, Shimadzu, Duisburg, Germany). Since the optical properties of AuNPs are affected by, among other things, particle size, UV–Vis was used as a proxy for observing clustering and declustering. Small AuNPs with a diameter below 10 nm have a low wavelength SPR band at 515–520 nm [18], whereas larger particles in the range 100–400 nm have a higher SPR band at > 650 nm [18]. Therefore, a bathochromic shift is indicative of the clustering of ultrasmall NPs; similarly, a hypochromic shift suggests declustering.

#### 2.3.2. Spectral Deconvolution

A spectral deconvolution script was written in the Python programming language to elucidate the UV–Vis absorbance spectra contributions from Rayleigh scattering, individual gold nanoparticles, and clusters. The script optimizes the input parameters of a model where Rayleigh scattering is fitted to an exponential decay [29] and gold nanoparticles/clusters are fitted to Voigt curves [30].

#### 2.3.3. Raman Spectroscopy

The Raman spectra of BPT-labelled constructs were captured using a system comprised of a 785 nm digital multi-mode spectrum-stabilized laser module (I0785MM0350MF-USB, IPS, Compton, CA, USA), a 785 nm high throughput probe (I0785P-FS-32-125-08, IPS, Compton, CA, USA), and a 750–1050 nm high performance smart spectrometer with a thermoelectric, cooled back-thinned CCD and a 25 μm slit (BTC655N, B&W Tek, Newark, DE, USA). One 5 μL drop of the Raman reporter-labelled sample in H_2_O was placed onto a steel slide, and spectra were recorded between −200 and 3500 cm^−1^.

#### 2.3.4. Raman Spectra Processing

A Raman spectra processing script was written in the Python programming language to smooth and flatten the raw data (Appendix A). A Savitzky–Golay (SG) filter [31] was applied to smooth each spectra, and an asymmetric least squares (ALS) smoothing algorithm [32] was used to approximate the baseline. The processing parameters were selected for the data on hand; typical values were a window size of 5 and an order of 3 for the SG filter and a smoothness of 1000 and an asymmetry of 0.01 for the ALS smoothing.

#### 2.3.5. Dynamic Light Scattering

Dynamic light scattering (DLS) measurements were performed on undiluted aqueous samples in plastic cuvettes at room temperature (25 °C), with the viscosity set to that of water. Refractive index and absorption were set to 0.20 and 3.32, respectively [33]. Measurements were performed on a Zetasizer Ultra (Malvern Panalytical, Malvern, UK) at least 3 times, and the average is reported.

#### 2.3.6. Transmission Electron Microscopy

Transmission electron microscopy (TEM) samples were prepared as follows. One drop of the undiluted sample in either H_2_O or chloroform (depending on the particle being analysed) was placed onto a carbon-coated copper grid and allowed to settle for 20 s; then, excess liquid was wicked away using filter paper. Imaging was performed with a CM120 TEM operating at 120 kV (Philips, Cambridge, MA, USA).

#### 2.3.7. Detection Specificity in a Biological Matrix (Plasma)

To demonstrate the detection specificity of the final construct in a biological matrix, the BPT-labelled clusters were diluted 10× into H_2_O or rabbit plasma, and the Raman spectra were recorded.

#### 2.3.8. Forced Degradation Study

The forced degradation of the constructs was carried out under conditions recommended by Blessy et al. [34]. Briefly, samples were stored in different stress conditions to assess degradation by acid hydrolysis (0.1 M HCl, 40/60 °C), base hydrolysis (0.1 M NaOH, 40/60 °C), and oxidation (3% *v*/*v* H_2_O_2_, 25/60 °C). UV–Vis spectral deconvolution data were used to assess stability by monitoring ultrasmall and cluster peaks.

#### 2.3.9. Glutathione Degradation Study

Glutathione (GSH) was used to assess the stability of the constructs in the presence of a physiological thiol compound. A GSH solution (1 mM; 0.3 mg mL^−1^) was prepared in PBS with the pH adjusted to 7.4 using Na_2_HPO_4_ (0.2 M, mqH_2_O). Constructs, obtained using the procedure described in Section 2.2.5, were washed three times with mqH_2_O and concentrated to 5× by centrifugation (2 × 10^3^ rpm for 10 min). An aliquot (10 μL) of concentrated constructs was added to the GSH solution (1 mM; 40 μL) for a final GSH concentration of 0.8 mM [35], and then the sample was incubated for 24 h at 37 °C with magnetic stirring; the UV–vis spectra were recorded.

## 3. Results

The UV–Vis spectra of ODA AuNPs show a sharp peak at 524 nm, indicative of small monodisperse AuNPs, which was also confirmed by TEM (Figure 2a). An analysis of ultrasmall particles (Figure 2a) using ImageJ (497 particles measured) revealed the particles to have an average diameter of 4.67 ± 1.74 nm (Appendix A). These particles were used to construct the clusters.

The extent of the bathochromic shift could be augmented by varying the concentration of either EGBMA or EtOH (Figure 3). Higher concentrations of EGBMA led to a greater degree of crosslinking and larger particles that, by the nature of gold nanoparticles, had a higher SPR wavelength. Increasing the concentration of EtOH made the solution more polar [36], which was likely the reason for the denser packing of hydrophobic ultrasmall AuNPs into the cluster, resulting in a greater number of neighbours per particle and intensifying the extent of the shift (Figure 3 and Appendix A). The most extreme shift observed in this set of experiments was to 710 nm with 1 µL mL^−1^ EGBMA and 10% *v*/*v* EtOH. The extent of the bathochromic shift obtainable by increasing the EGBMA concentration plateaued at 1 µL mL^−1^, so further increases would have had a negligible effect. However, in the case of EtOH, the shift continued to increase up the highest tested concentration, suggesting that further optimization is possible should a stronger shift be required.

GCPQ was used to transfer the EGBMA-clustered AuNPs to the aqueous phase by way of a thin film approach. The resulting particles were analysed by UV–Vis spectroscopy (Figure 4a). Various GCPQ concentrations were tested in the range of 0.1–1 mg mL^−1^. There was a significant increase in absorbance as the concentration of GCPQ increased, indicative of higher cluster encapsulation. The range from 1 to 10 mg mL^−1^ was selected for the further optimization of particle size and dispersity. GCPQ concentrations of 2, 4, 6, 8, and 10 mg mL^−1^ were tested and analysed by DLS (Figure 4b and Appendix A). A weak positive trend was observed between GCPQ concentration and particle diameter; as the concentration increased from 2 to 10 mg mL^−1^, particle diameter increased from 254 to 278 nm. However, the increase did not have a significant impact on the final construct, as all particles were within the target size range. No correlation was observed between GCPQ concentration and polydispersity, with all analysed particles showing narrow PDIs of 0.06–0.15. A mid-range GCPQ concentration of 5 mg/mL was selected for the final construct.

To optimize labelling, BPT was added at various stages of clustering—30 min before, concurrently, or 30 min after the addition of EGBMA, the clusters were coated with GCPQ and analysed by TEM and Raman spectroscopy (Figure 5). Adding BPT before EGBMA led to less densely packed AuNPs, probably due to the BPT occupying a large fraction of the AuNP surface area and leaving less opportunities for EGBMA to crosslink particles. Adding BPT after clustering produced particles that appeared less spherical, which could have been due to a second stage of aggregation induced by the BPT (Appendix A). Concurrent addition yielded the most similar particles to the original BPT-free clusters in terms of core density and sphericity. All particles appeared to aggregate on TEM images (Figure 5a, Appendix A), though this was most likely an artefact of the drying process because such structures would have been observed by DLS. To see freely dispersed particles, it would be necessary to employ cryo-TEM.

As the concentration of BPT in the labelling reaction increased, the amount of BPT on the construct’s surface—and therefore the Raman intensity—increased (Figure 5b). The relationship followed a logarithmic trend (intensity = 735.18ln([BPT]) + 2558.1, R^2^ = 0.9819) suggesting that for the clusters used in this experiment, the surface coverage was approaching saturation.

Raman spectra were obtained for the constructs in the presence and absence of rabbit plasma to demonstrate the detection specificity in biological matrix (Figure 6). Though the sensitivity was reduced in plasma vs. H_2_O (Figure 6a)—a problem that can be overcome by increased integration times, for example—the specificity was unchanged (Figure 6b), showing that the Raman scattering from the biological matrix was distinguishable from that of the constructs. The specificity of the RaNT construct is demonstrated by these data.

The deconvolution of UV–Vis spectra obtained from samples stored in both aqueous and acidic (0.1 M HCl) conditions showed no sign of degradation at either 40 or 60 °C, with the cluster peak remaining dominant over the ultrasmall peak. Samples stored under oxidizing (3% *v*/*v* H_2_O_2_) conditions remained stable at 25 °C; however, at 60 °C, the intensity of absorbance dramatically decreased. Under basic (0.1 M NaOH) conditions, the deconvoluted spectrum displayed a significant shift from the cluster peak to the ultrasmall peak (Figure 2d), with the effect being more pronounced at 60 °C (Appendix A). Declustering was also evident in the TEM analysis (Figure 2).

Constructs were incubated under physiological conditions (37 °C, pH 7.4, [GSH] ≈ 0.8 mM [35]) to test the hypothesis that one potential degradation pathway in vivo may occur via interaction with biological thiols. It was observed that over the course of 24 h, the constructs underwent significant declustering (Figure 7).

## 4. Discussion

The target RaNT architecture shown in Figure 1 was successfully prepared, and all its key characteristics—namely a reversible ultrasmall-in-nano architecture allowing for both the potential excitability of sub-5 nm particles and the optical properties of >100 nm particles, a coating with a highly tuneable and modifiable biopolymer to enable dispersion within aqueous media, and Raman labelling showing specific detection in biological media—were demonstrated.

### 4.1. Construct Synthesis and Optimization

We synthesized ultrasmall gold nanoparticles with an average diameter of 4.67 ± 1.74 nm (Appendix A), which should be excretable upon declustering [22,23]. EGBMA was found to cluster the ultrasmall particles in a way that elicited a bathochromic shift (Figure 2). For interparticle coupling to have a significant effect on the SPR of a construct, the centre-to-centre distance (d) must be less than 5× the individual particle radius (R) [37]. Therefore, in the case of 5 nm particles, d must be less than 12.5 nm, and the surface-to-surface distance must be less than 7.5 nm; otherwise, the shift is negligible. Furthermore, the closer the particles are, the stronger this coupling effect [38,39]. Therefore, it is important to select a linker, such as EGBMA, that is short enough (molecular size ≈ 12.3 Å, thiol-to-thiol; Appendix A) to bring particles into close proximity.

The intensity of Raman signal measured in the aqueous media was related to the time of addition of BPT as follows: BPT before EGBMA > BPT with EGBMA > BPT after EGBMA. This was probably due to the decreased opportunity for the Raman reporter molecule to situate itself in the hotspots [40] between gold nanoparticles with the later addition (Raman spectra and TEM images can be found in Appendix A).

The measured Raman intensity of BPT-labelled increased as a logarithmic function of BPT concentration, approaching a plateau that suggested a saturation of the gold surface. To further increase the Raman response, the construct surface area available for BPT labelling could be increased. This could be achieved by altering the addition time point of BPT, as outlined above, or by utilizing different cluster densities via the modulation of EGBMA and EtOH concentrations during clustering. Despite the relative flatness of the concentration Raman response curve, these data essentially demonstrate a specific and sensitive SERS response.

### 4.2. Construct Characterization

Constructs were found to be stable under acidic (40 or 60 °C), and room temperature oxidizing conditions. However, they were found to precipitate under accelerated (60 °C) oxidizing conditions. This was potentially due to the homolytic cleavage of the HO–OH bond liberating the hydroxyl radical (2HO•) at elevated temperatures [41]. The hydroxyl radical dominated the observed degradation, which is likely indicative of the degradation of the GCPQ coating [42], leaving bare hydrophobic clusters that precipitated in the aqueous solution (Appendix A). Constructs demonstrated declustering under basic (40 or 60 °C) conditions, with the declustering more rapidly occurring at the higher temperature. This finding suggests that the labile linkers are undergoing hydrolysis, as hypothesized. This demonstrates a key feature of the construct—the ability to liberate sub-5 nm AuNPs after clustering. Speculatively, the alkaline-instability of the construct may lead to preferential accumulation in the acidic microenvironment (pH 6.3–7.0) of a tumour [43,44], as well as the degradation of constructs that reside at physiological pH, though further studies are required to demonstrate this. Though declustering is essential for the potential excretion of the ultrasmall particles, it will be necessary to show that this does not prematurely occur during storage. This risk has been significantly reduced by showing that the GCPQ-coated constructs can be freeze-dried for storage and reconstituted without changing their optical properties.

Though the forced degradation of the constructs provides insights into the degradation pathways and demonstrates the ability of the constructs to revert to sub-5 nm AuNPs under stress conditions, it does not show how the constructs will behave under physiological conditions. We hypothesize that one potential degradation pathway in vivo may be via interaction with biological thiols. GSH is the most abundant low molecular weight biological thiol [45] and was therefore selected to test this hypothesis. Over the course of 24 h under physiological conditions (37 °C, pH 7.4, [GSH] ≈ 0.8 mM [35]), the constructs underwent significant declustering (Figure 7). This is the first time that biological thiols have been demonstrated to play a potential role in the declustering of an ultrasmall-in-nano construct. Similar studies have been conducted with higher GSH concentrations and without pH adjustment (i.e., acidic conditions), and only slight declustering was detected (Appendix A), which hints at the mechanism of the observed disaggregation. The thiol of GSH likely bonds with the surface of the AuNPs, generating carboxylic-acid-coated particles that are only be dispersible in neutral/basic conditions [46]. This initial proof of concept will be followed by work to determine the declustering kinetics.

## 5. Conclusions

Ultrasmall-in-nano Raman NanoTheranostic (RaNT) constructs were generated with desired optical and structural properties. The final gold constructs were synthesized using octadecyl amine, a labile dithiol linker, and GCPQ as capping, clustering, and phase transfer agents, respectively. The final construct was a ~250 nm cluster, with strong absorbance in the phototherapeutic window, that was capable of producing a specific Raman response in a biological matrix and able to revert to sub-5 nm AuNPs under simulated physiological conditions. Various aspects of this construct have previously been demonstrated in the literature, including an ultrasmall-in-nano architecture [47,48,49] prepared from ultrasmall nanoparticles and the use of AuNPs for SERS probes [50,51]; however, this is the first time that a Raman NanoTheranostic was prepared while combining these features with the proven ability to disaggregate into potentially excretable nanoparticles.

## Figures and Tables

**Figure 1 nanomaterials-11-02181-f001:**
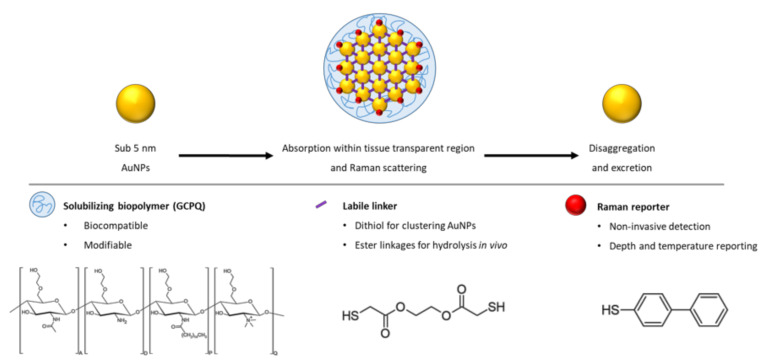
Outline of the Raman NanoTheranostic architecture.

**Figure 2 nanomaterials-11-02181-f002:**
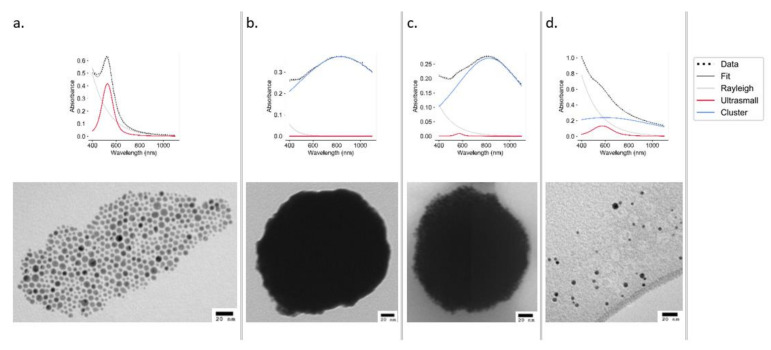
UV–Vis spectra (top panels) and TEM images (bottom panels) of AuNPs at stages of synthesis and declustering: ultrasmall ODA AuNPs (**a**), EGBMA clusters (**b**), GCPQ-coated clusters (**c**), and ultrasmall AuNPs after declustering using 0.1 M NaOH (**d**).

**Figure 3 nanomaterials-11-02181-f003:**
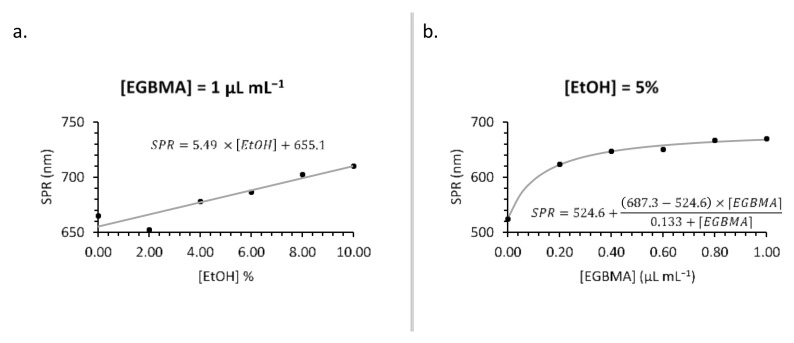
Synthesis optimization: effect of [EtOH] (**a**) and [EGBMA] (**b**) on particle SPR wavelength.

**Figure 4 nanomaterials-11-02181-f004:**
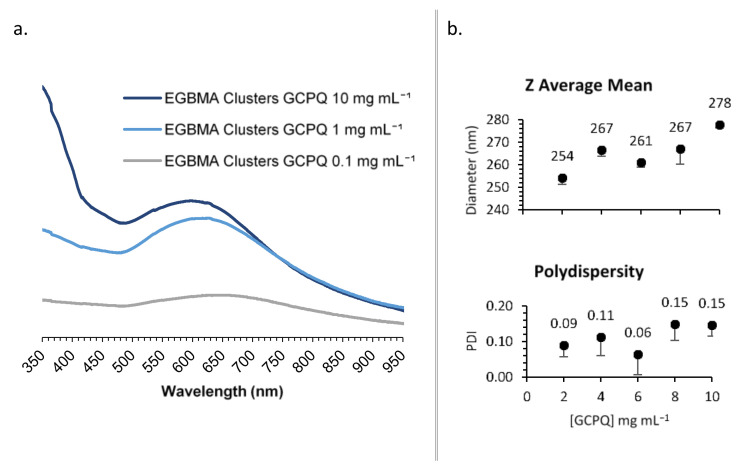
Synthesis optimization: effect of [GCPQ] on UV–Vis spectra (**a**); DLS data (**b**).

**Figure 5 nanomaterials-11-02181-f005:**
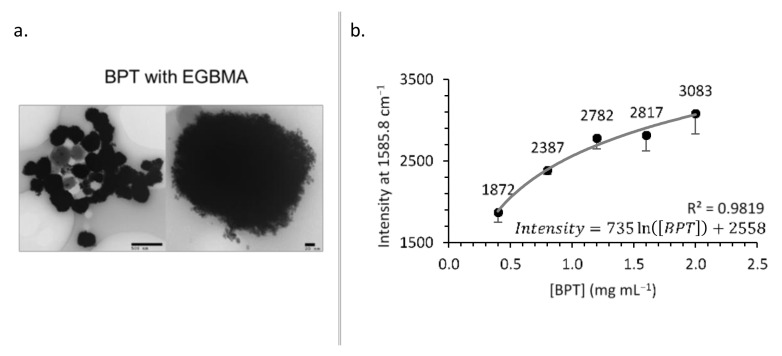
BPT-labelled clusters: TEM images (**a**) and effect of [BPT] on Raman intensity (**b**).

**Figure 6 nanomaterials-11-02181-f006:**
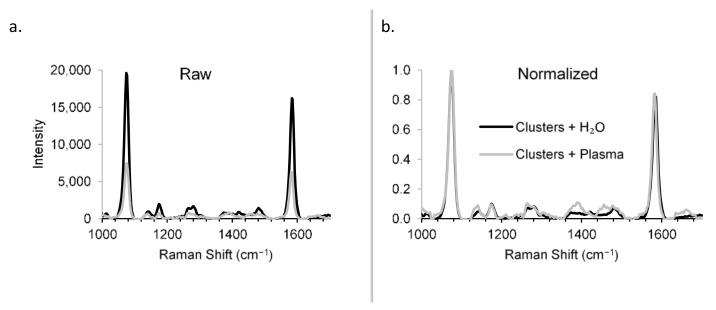
Detection specificity in biological matrix. Raman spectra of BPT-labelled clusters in H_2_O and plasma: raw (**a**) and normalized (**b**).

**Figure 7 nanomaterials-11-02181-f007:**
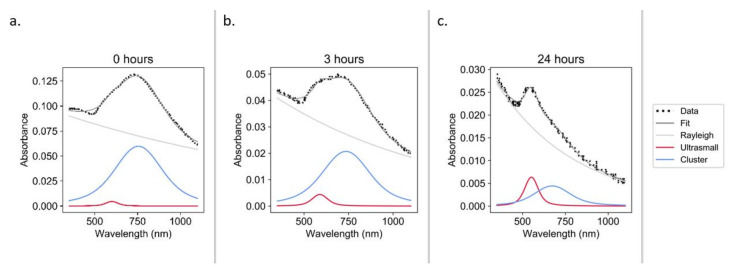
UV–Vis spectra during the glutathione degradation of constructs in PBS at pH 7.4: 0 h (**a**), 3 h (**b**), and 24 h (**c**).

## Data Availability

Data is contained within the article or Appendix A.

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
