# Peer review of "Development of Bio-Functionalized, Raman Responsive, and Potentially Excretable Gold Nanoclusters"

_nanomaterials, 2021, doi:10.3390/nano11092181_

Round 1

Reviewer 1 Report

I recommend the paper 1352352 considered for publication in the Nanomaterials in the current version.

Author Response

No comments to address.

Reviewer 2 Report

The manuscript is a clear and convincing description of  the experiments  which demonstrate the possibility of clustering / declustering of  encapped gold nanoparticles. The procedure may provide an efficient  tool for Raman - nanotheranostic application. 

Author Response

No comments to address.

Reviewer 3 Report

In this manuscript, Mellor et al., reported the synthesis and covalently cross-linked gold nanoparticle aggregates for enhances SERS detection.

The authors prepared octadecylamine (ODA) capped polydisperse nanoparticles using sodium borohydride reduction. The nanoparticles were then treated with Biphenyl-4-thiol which is sensitive for SERS.  Further, the ODA capped nanoparticles were treated with ethylene glycol bis acrylamide (EGBAM), which resulted in cross-linked mediated aggregation into larger structures. The nanoparticles and the aggregates were characterised using TEM and DLS. Finally, the structures can be disassembled under simulated physiological conditions.

Corrections and comments:

  1. The title is misleading as the authors state "potentially excitable  gold nanoclusters". Since there is no experimental evidence for excretion, and gold nanoclusters refers to "atomically precise gold nanoparticles" This might mislead the authors.  May be the title should be changed to ....raman responsive nanoparticle aggregates.
  2. The nanoparticles that the authors have used is highly polydisperse as evident based on TEM. Please provide a size distribution analysis and histogram.
  3. There is no low mag image of cross-linked aggregates as the authors show only one particles. A size distribution along with low mag images with multiple particles will be useful.
  4. There is no DLS data provided in the manuscript
  5. Please pay attention to subscripts for example NaBHin the entire manuscript.
  6. The authors state that " sub-5nm gold nanoparticles formed into a 71 reversible 100-400 nm cluster".  It is not reversible as one could only disassemble and not able to assemble again.  
  7. The authors state that " Unfortunately, particles which  exhibit this SPR band tend to be in the size range of 100-400 nm; a particle size that  would not be sufficiently excreted by the kidneys". However, it is well known that SPR is observed in particles of much smaller size. For example gold nanoparticles and gold nano rods of 30 nm.  See: https://onlinelibrary.wiley.com/doi/epdf/10.1002/anie.201802420 
